# Acute Effects of Short Static, Dynamic, and Contract–Relax with Antagonist Contraction Stretch Modalities on Vertical Jump Height and Flexibility

**DOI:** 10.3390/sports13040115

**Published:** 2025-04-10

**Authors:** Clément Cheurlin, Carole Cometti, Jihane Mrabet, Jules Opplert, Nicolas Babault

**Affiliations:** 1Centre d’expertise de la Performance, Faculté des Sciences du Sport, Université Bourgogne Europe, 3 Allée des Stades Universitaires, BP27877, 21078 Dijon, France; clement.cheurlin@hotmail.fr (C.C.); carole.cometti@u-bourgogne.fr (C.C.); jiha.mrabet@gmail.com (J.M.); opplert.jules@gmail.com (J.O.); 2Maison de Santé, 50 Place du Château de Randens, 73270 Beaufort, France; 3INSERM U1093, CAPS, Faculté des Sciences du Sport, Université Bourgogne Europe, 3 Allée des Stades Universitaires, BP27877, 21078 Dijon, France; 4E-Motion Coaching, 1000 Brussels, Belgium; 5Research Department, Medinetic Learning, 4 rue Marceau, 75008 Paris, France

**Keywords:** countermovement jump, proprioceptive neuromuscular facilitation, warm-up, short-duration stretch

## Abstract

The present study investigated the acute effects of different stretching modalities applied within a warm-up on flexibility and vertical jump height. Thirty-seven young adults participated in four randomized experimental sessions, each corresponding to a different condition: static stretch (SS), dynamic stretch (DS), contract–relax with antagonist contraction (CRAC) or a control condition with no stretch (CTRL). Conditions were five min in total duration, including 2 × 15 s stretches for each muscle group (knee flexor, knee extensor, and plantar flexor muscles). Ten min and five min of cycling preceded and followed these procedures, respectively. Hamstring flexibility and a series of countermovement jump (CMJ) measurements were interspersed within this procedure. Except for CTRL, hamstring flexibility significantly increased (*p* < 0.01) after all experimental procedures (7.5 ± 6.6%, 4.1 ± 4.9%, and 2.7 ± 6.0% for CRA, SS, and DS, respectively). The relative increase was significantly greater for CRAC as compared CTRL (*p* < 0.001). Vertical jump height significantly decreased (*p* < 0.05) immediately after SS (−2.3 ± 3.9%), CTRL (−2.3 ± 3.5%), and CRAC (−3.2 ± 3.3%). Jump height was unchanged after DS (0.4 ± 4.5%). Whatever the condition, no additional jump height alteration was obtained after the re-warm-up. The main findings of the present study revealed that DS is more appropriate for maintaining vertical jump height. However, stretching has no major effect when performed within a warm-up. In contrast, if the main objective is to increase flexibility, CRAC is recommended.

## 1. Introduction

Stretching is a well-known technique generally applied for therapeutic usage, relaxation, and for warm-up routines in most physical activity programs. Various stretching modalities could be used. Static, dynamic, or proprioceptive neuromuscular facilitation (PNF) techniques are frequently used [1] in sports settings. All of these techniques have various advantages, and many parameters should be controlled for optimal effects.

Static stretching (SS) implies the maintenance of a stretched position to the maximal point of discomfort to enhance range of motion (ROM), mostly by a reduction in muscle stiffness [2]. Dynamic stretching (DS) incorporates whole-body or analytic movements and involves actively and rhythmically contracting a muscle group through parts of its functional ROM. This permits individuals to elevate core temperature, enhance motor unit excitability, improve kinesthetic awareness, and maximize ROM [3]. PNF techniques offer a wide range of benefits but require rigorous procedures usually applied by professionals such as physiotherapists [4]. Numerous investigators found that PNF techniques produced the largest gains in muscle flexibility as compared to other forms of stretching [2,5,6]. However, contradictory results were often obtained [7,8]. Therefore, although flexibility is the primary objective of stretching, evidence of the best modality are still lacking [9,10,11].

In addition to the likely gains in ROM, acutely induced effects of stretching on muscle performance are often examined isolated or in combination with dynamic exercises [12]. Each stretching modality has already been tested on various outcomes such as power, agility, sprint time or vertical jump height [13,14,15]. However, very few studies have compared more than two different techniques in a single study and even fewer on a specific muscle performance outcome such as vertical jump height. For instance, authors found detrimental effects of stretching on countermovement jump (CMJ) height [16]. However, ballistic stretching resulted in a smaller decrease in vertical jump height than both SS and PNF [16].

Moreover, PNF could be conducted with various modalities. Among them, contract–relax with antagonist contraction (CRAC) remains little explored. Authors showed CRAC significantly decreased vertical jump performance as compared to SS and a general warm-up composed of ten body-weight dynamic exercises [14]. Results also indicated a non-significant difference in flexibility pre- and post-treatment. Despite the widespread use of all of these various stretching techniques, no study has compared CRAC to the more frequently used SS and DS on both vertical jump performance and flexibility. Therefore, the present study aimed to determine the acute effects of SS, DS, and CRAC stretching modalities on the flexibility of lower limb muscles and vertical jump height. We hypothesized that CRAC will be more efficient for flexibility while DS will be more efficient for vertical jump height.

## 2. Materials and Methods

### 2.1. Participants

A total of 37 young, healthy adults (14 females and 23 males) voluntarily participated in this study (age: 20.3 ± 1.1 years, mass: 57.7 ± 8.5 kg, height: 163.1 ± 6.5 cm for women and age: 20.7 ± 1.5 years, mass: 70.5 ± 8.1 kg, height: 178.9 ± 6.0 cm for men). All were physically active with at least two training sessions per week. The mean training volume was 4.3 ± 4.4 h per week. Volunteers were predominantly track and field or team sport athletes. Each participant read and signed an informed consent form outlining the experimental procedure. The study was conducted according to the Declaration of Helsinki, approved by the local committee on human research and following ethical standards. Volunteers who had experienced a lower limb trauma in the four months preceding the experiment or who were suffering from a rheumatic pathology were not included. One week before all testing sessions, volunteers performed a familiarization session focused on instructions for countermovement jumps (CMJs) and stretch procedures. The sample size was calculated a priori using G*Power (version 3.1.9.6, free software available at https://www.psychologie.hhu.de/arbeitsgruppen/allgemeine-psychologie-und-arbeitspsychologie/gpower, accessed on 6 April 2025.) with the following parameters: medium effect size of 0.25, power of 0.8, probability error of 0.05, and correlation among repeated measures of 0.7. A minimal sample size of 32 individuals was proposed.

### 2.2. Experimental Design

This study was designed to determine the acute effects of different stretching modalities on flexibility and vertical jump height. Participants were tested during four different and randomly presented sessions. The sessions were separated by at least 72 h and were all scheduled at the same time of day. One session was used as control (CTRL, no stretch) and the other three were dedicated to stretching (SS, DS, or CRAC). Stretching was conducted targeting knee flexor, knee extensor, and plantar flexor muscles. Two 15 s stretches were performed for each muscle group. Before and after stretching, participants were requested to conduct a warm-up exercise that was composed of light pedaling. Hamstring flexibility and CMJ height were quantified within this procedure. The experimental design is shown on Figure 1.

A passive flexibility assessment of hamstring muscles was first performed with an inclinometer (Dr. Rippstein^®^, Zurich, Switzerland) positioned on the right quadriceps. Volunteers were lying supine and were asked to relax. The experimenter blocked the left leg to keep it straight and flat on the ground and lifted the right leg to the point of discomfort while keeping it straight. The lower limb angle with respect to the ground was then taken as pre-experimental ROM—an index of initial flexibility.

Each session then consisted of a standardized 10 min warm-up on a cycle ergometer (Technogym, Gambettola, Italy). The power was set at 70 W and 80 W for women and men, respectively (70–80 revolutions per minute). Immediately after this cycling warm-up, a first series of two CMJs was performed (CMJ Post-WU). After one minute of rest, one of the following conditions was randomly performed: (i) control (CTRL, no stretch), (ii) contract–relax with antagonist contraction stretching (CRAC), (iii) dynamic stretching (DS), or (iv) static stretching (SS). These interventions were five minutes long. Immediately after stretching, a second series of CMJs was conducted (CMJ Post-S). Then, volunteers rested for 2 minutes and performed 5 min of cycling, as a re-warm-up, at a higher intensity (100 W and 125 W power output for women and men, respectively; 90–100 revolutions per minute). This re-warm-up was immediately followed by the last series of jumps (CMJ Post-Re-WU), which was immediately followed by the measurement of post-experimental passive flexibility with the same procedure as shown above.

### 2.3. Vertical Jump Performance

Vertical jump performance was assessed using a portable force plate (Quattro Jump; Kistler, Winterthur, Switzerland) measuring vertical reaction forces. CMJs were performed starting from a standing position, then squatting down to a 90° knee angle (±5°), and then extending the knees in one continuous movement. During the CMJs, arms were kept on the hips to minimize their contribution. Vertical jump height was calculated by the force plate software. The best performance of the two trials of each series was used for analyses.

### 2.4. Stretching Protocol

The different stretching modalities followed similar protocols, with characteristics presented in Table 1. Each muscle group was stretched twice and each repetition lasted 15 s. A sequential order was applied: hamstrings first, then quadriceps, and finally plantar flexors, always starting with the left side. No rest period was allowed when switching limbs. For the CTRL session, volunteers remained seated for five minutes.

### 2.5. Statistical Analysis

Statistical analyses were conducted using JASP (version 0.14, JASP Team 2020, University of Amsterdam, freely available at https://jasp-stats.org/download/, accessed on 17 February 2024). First, the normality and sphericity were tested and confirmed by the Shapiro–Wilk and Mauchly's tests, respectively. Then, a three-way ANOVA with repeated measures was conducted in order to compare the main effects or interactions of sex × modality × time interactions. Modality refers to SS vs. DS vs. CRAC vs. CTRL. Time refers to the comparison between pre and post hamstring flexibility, and CMJ Post-WU, CMJ Post-S, and CMJ Post-Re-WU for vertical jumps. The relative changes for hamstring flexibility and vertical jumps were also compared by means of the ANOVA. If the sphericity was not verified, a Greenhouse–Geisser correction was performed. In cases of significant effects, post hoc tests with Bonferroni corrections were conducted. The partial eta square (pη2) was calculated from the ANOVA. Thresholds of 0.01, 0.06, and above 0.14 represented small, medium, and large differences, respectively [17]. Subsequently, Cohen’s d was calculated with values <0.5, 0.5–1.2, and >1.2 representing small, medium, and large magnitudes of change, respectively [17]. *p* < 0.05 was taken as the threshold for statistical significance and results are presented as mean values ± SD.

## 3. Results

Firstly, statistics demonstrated similar initial values for all four experimental sessions for flexibility and vertical jump. Secondly, statistical analyses were conducted including a sex factor. Except a larger flexibility (*p* < 0.001, pη2 = 0.270, large) and lower vertical jump height (*p* < 0.001, pη2 = 0.280, large) for women as compared to men, no sex effect was observed for flexibility and vertical jump changes after warm-up and stretching interventions.

A significant interaction (modality × time) was also found for hamstring flexibility (*p* < 0.001, pη2 = 0.205, large). Flexibility was significantly improved after SS (*p* < 0.001, d = 0.211, small) and CRAC (*p* < 0.001, d = 0.381, small) (Table 2). Relative variations (*p* < 0.001, pη2 = 0.152, large) demonstrated significant larger flexibility increases following the CRAC intervention as compared to CTRL and DS (*p* < 0.001, d = 1.043, medium; *p* = 0.002, d = 0.809, medium, respectively) (Figure 2). CRAC compared to SS was close to the statistical significance (*p* = 0.054, d = 0.579, medium). No difference was observed between SS, CTRL, and DS.

A significant interaction (modality × time) was found for vertical jump height (*p* = 0.018, pη2 = 0.076, medium). Post hoc analyses showed that the CMJ Post-S was significantly lower after CTRL (*p* = 0.033, d = 0.130, small), SS (*p* = 0.001, d = 0.029, small) and CRAC (*p* < 0.001, d = 0.171, small) as compared to CMJ Post-WU (Table 2). No difference was observed for DS whatever the time point. Relative variations (*p* < 0.001, pη2 = 0.152, large) demonstrated significant larger CMJ reductions following CTRL, SS, and CRAC stretching as compared to DS (*p* = 0.012, d = 0.702, medium; *p* = 0.013, d = 0.697, medium; and *p* < 0.001, d = 0.917, medium, respectively) (Figure 3). Conducting the re-warm-up allowed values to return to baseline (no significant difference between Post-WU and Post-Re-WU) for all four conditions.

## 4. Discussion

The present study aimed to compare the acute effects of static, dynamic, and CRAC stretching modalities on hamstring flexibility and vertical jump height. Our results partly confirmed our a priori hypothesis. We observed a significant ROM increase with all stretch modalities but even more with CRAC treatment. For vertical jump performance, DS is recommended since this intervention did not change nor impede CMJ height, whilst SS, CRAC, or no stretching were detrimental. However, performing a light re-warm-up after stretching permitted values to return to baseline.

Gaining flexibility is the primary objective of stretching. Our results confirmed stretching interventions improved flexibility. Our results also revealed CRAC (a form of PNF) was more effective than DS and close to statistical significance as compared to SS. Previously, CRAC has been found to be more efficient than SS [18]. In contrast, others were unable to find any flexibility change with this modality applied 3 × 30 s to hamstring and quadricep muscles [14]. These conflicting results could be attributed to the different durations used and to the isometric contractions performed during their stretching protocol. In fact, long PNF durations could produce fatigue and therefore reduce stretching efficiency. The short CRAC stretch applied here (2 × 15 s) therefore seemed more effective.

One can question the effectiveness of the different PNF techniques for ROM increases. However, very few studies compared these different modalities. For instance, authors revealed the hold–relax and contract–relax PNF modalities have similar acute effects [19]. The inconsistency in the literature is often observed for all PNF techniques. While some studies have found that PNF was the most efficient technique to gain flexibility in different populations [5,6,20,21,22], other studies and recent meta-analyses were unable to find differences between PNF and SS [8,9,23,24,25]. Moreover, as pointed out in the literature [26], small effect sizes are often noted between pre and post intervention changes. However, such conclusions could be mitigated here, since medium effects are obtained when comparing relative changes between modalities. Moreover, although not significant, effect sizes revealed a medium effect when comparing CRAC and SS. The apparent greater efficiency of CRAC is unclear but could partly be attributed to neural aspects such as autogenic or reciprocal inhibition that may favor muscle relaxation [27]. Muscle–tendon properties could also be involved. PNF seemed to produce changes in muscle and tendon stiffness while SS mostly decreased muscle stiffness [2]. Also, in contrast with SS and DS, CRAC was applied by an experimenter which could increase this stretching modality’s efficiency. However, given the poor literature (as opposed to the abundant SS or DS studies), studies should focus on these PNF techniques to determine physiological mechanisms and acute/chronic effects. For instance, while short-duration CRAC treatments are recommended here (to avoid fatigue), the intensity of the contractions should also be questioned. For example, authors often used sub-maximal contractions during PNF techniques to limit fatigue and reduce a likely injury risk [28].

Moreover, flexibility was only improved after SS and CRAC. The lack of flexibility changes after DS contradicts the literature [26,29]. However, one should remember the short stretching duration used here (two repetitions of 15 s). This short duration could mask a likely increase in flexibility with longer stretch duration. Indeed, the longer the stretch duration, the larger the gains in range of motion [30,31].

In our study, vertical jumps were performed at different time points: after an initial warm-up, after stretching procedures, and after re-warm-up. Except for DS, our study revealed impairments immediately after stretching. A re-warm-up consisting of light pedaling permitted values to return to baseline and to alleviate potential negative stretching effects.

Previous studies have investigated stretching-induced effects during power- and explosive-type actions in different populations [22,32,33,34,35]. Many of these have shown similar conclusions to ours with respect to the lack of impairments following DS [16,34,35,36,37]. Also, the decrease in vertical jump height after SS was similar to that observed in other studies [16,36]. A decrease in vertical jump height after PNF modalities is also often registered [14,16,22].

The decrease in vertical jump height after a single stretching session is generally attributed to various mechanisms. Authors previously registered reductions in voluntary activation or electromyographic activity [28]. Also, mechanical origins such as alteration of the elastic properties of the muscle–tendon complex have been measured [12,38]. The decreased stiffness of the muscle–tendon complex could impair force production in muscles as a result of changes in their force–velocity and length–tension relationships [39]. The vertical jump performance being correlated to both the rate of force development [40] and stiffness [41], and the likely musculotendinous unit stiffness decrease induced by stretching routines may both lower CMJ height.

The lack of CMJ height decrease after DS has often been observed [33,42,43]. This contrasting effect as compared to SS or CRAC is generally attributed to three likely mechanisms, including increased temperature [44], more sport-specific movements [45], and altered central drive [28]. To confirm these hypotheses, authors have demonstrated DS effects resulting from both a muscle–tendon stretch and a muscle warm-up [3]. The muscle–tendon stretching would partly counteract muscle warm-up effects. This hypothesis is partly confirmed by our results, considering the re-warm-up after stretching. Indeed, for all experimental conditions, values were back to baseline after the re-warm-up. The finding that the negative effects of stretching on performance are minimized or restored when followed by a conditioning activity, composed of maximal contractions (plyometric or concentric contractions), is also well documented [12,46]. Accordingly, both high-intensity contractions or long-duration low-intensity efforts (with reduced fatigue) may be alternatives to minimize stretching-induced force decrease.

The present study has several limitations. First, flexibility was only quantified in hamstring muscles. Because stretching was conducted on knee flexor, knee extensor, and plantar flexor muscles, an evaluation of flexibility, or more generally speaking on stiffness, would be of interest in order to determine some possible muscle dependencies on various stretching modalities. Indeed, stretching effects have previously been shown to be muscle-group-dependent [38]. Second, vertical performance was only quantified using height. Quantifying the rate of force development during such functional tests would help researchers to understand the relationship between likely stiffness alterations and mechanical output. Third, although the present study included warm-up or re-warm-up, a more comprehensive warm-up procedure should be used in order to generalize the present results to a real-world warm-up routine. Finally, CRAC was used as a PNF stretching modality. As for static or dynamic stretching, various parameters could influence our findings. The manipulation of either or both contraction or stretching durations and intensities should therefore be explored. In addition, future studies should explore in depth the physiological mechanisms behind the different stretching modalities.

## 5. Conclusions

The results of the present study reveal that all stretching modalities are interesting for hamstring flexibility improvements, with a more pronounced effect of CRAC. Moreover, SS and CRAC immediately produced a reduction in vertical jump performance, whereas it was unaffected after DS. Consequently, CRAC or SS prior to an explosive athletic movement are not recommended. However, whatever the stretching modality, performing a low-intensity, long-duration re-warm-up seemed to alleviate potential negative acute effects of stretching. CRAC is therefore recommended for those who expect a large gain in flexibility. In contrast, DS is recommended within a warm-up routine to avoid vertical jump impairments.

## Figures and Tables

**Figure 1 sports-13-00115-f001:**
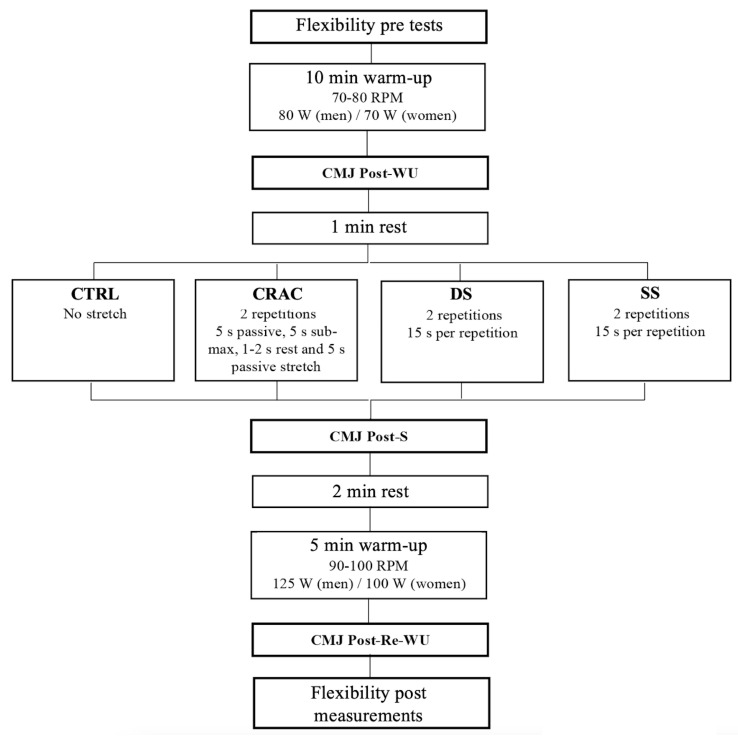
A schematic view of the experimental design. PRE: pre-experimental; RPM: revolutions per minute; CMJ: countermovement jump; CTRL: control condition; CRAC: contract–relax–antagonist-contract stretching; DS: dynamic stretching; SS: static stretching; POST: post-experimental.

**Figure 2 sports-13-00115-f002:**
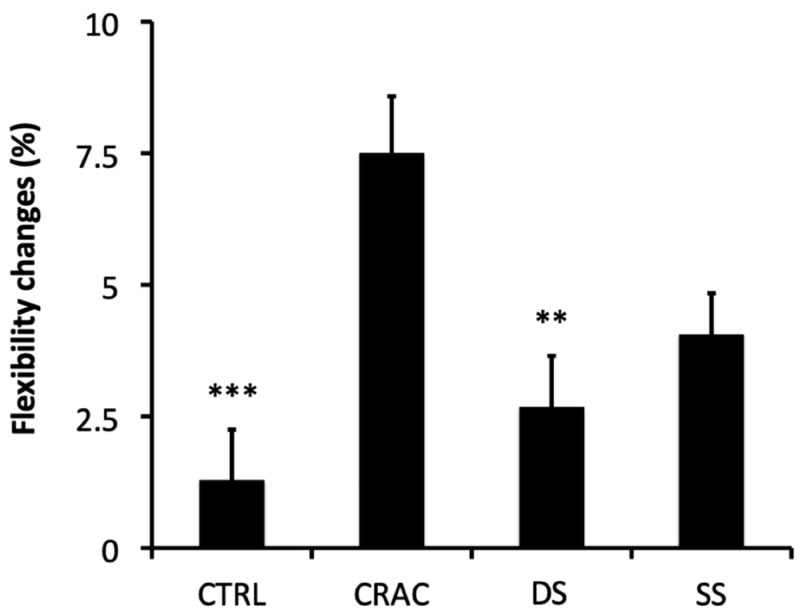
Percentage changes (mean values ± SD) of hamstring flexibility for all experimental conditions. Significant difference as compared to CRAC are shown (**: *p* < 0.01 and ***: *p* < 0.001).

**Figure 3 sports-13-00115-f003:**
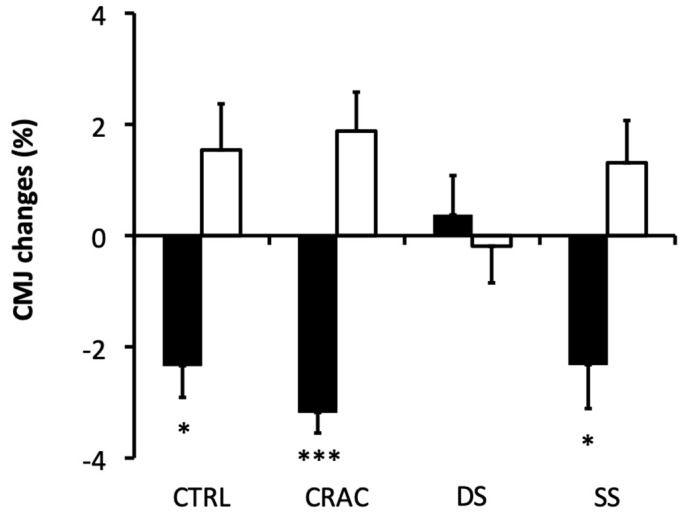
Percent changes (mean values ± SD) in vertical jump height for the different experimental conditions. Values are expressed relative to the Post-WU condition. Black bars are post-stretching (Post-S). White bars are post re-warm-up (Post-Re-WU). Significant differences as compared to Post-WU and as compared to the DS condition are shown (*: *p* < 0.051 and ***: *p* < 0.001).

**Table 1 sports-13-00115-t001:** Description of stretching routines.

Stretch	Procedure	Hamstrings	Quadriceps	Plantar Flexors
CRAC	5 s passive stretch followed by 5 s sub-maximal voluntary concentric contraction, 1–2 s rest, and 5 s passive stretch to point of discomfort concomitant with maximal concentric contraction of antagonist muscle group	In a lying position with extended leg, passive stretching of the hip flexors was performed by the experimenter, followed by hamstring contraction, a rest period and a second passive stretch coupled with knee extensions.	Lying on the side with extended hip, subjects passively stretched the knee extensors, followed by contraction of the same muscle. After a rest period, they performed a passive stretch coupled with hamstring contraction.	In a lying position, passive stretching of plantar flexors with the tibial anterior muscle was performed, followed by concentric contraction of target muscle, a rest period, and another stretch by the experimenter coupled with plantar flexor contraction.
DS	2 repetitions per muscle group: 15 movements at 1 Hz until maximal range of motion	The subjects contracted the hip flexors with the knee extended and flexed the hip joint so that the leg swings forward.	The subjects contracted the hamstrings and flexed the knee joint so that the heels hit the buttocks.	The subjects raised one foot from the floor and fully extended the knee. Then, subjects contracted the dorsiflexors so that foot/toes point upward.
SS	2 repetitions per muscle group: 15 s passive static stretching until the point of discomfort	From a standing position, the subject placed their heel on a Swedish bench (45 cm high) and repositioned their pelvis backwards, keeping their torso erect.	From a standing position and keeping torso erect, the subject bent one knee and brought the heel up towards the buttock; the foot was held with their hands.	In a stride-stand position, with back leg straight and forward leg slightly bent, the subject leaned forward with both hands against a wall.

CRAC, contract–relax with antagonist contraction; DS, dynamic stretching; SS, static stretching.

**Table 2 sports-13-00115-t002:** Vertical jump height and hamstring flexibility during the experimental protocol.

Test	CTRL	CRAC	DS	SS
Vertical jump height (cm)				
CMJ Post-WU	43.3 ± 8.3	43.1 ± 8.0	43.0 ± 8.3	43.3 ± 8.6
CMJ Post-S	42.3 ± 8.0 *	41.7 ± 7.7 *	43.1 ± 7.9	42.2 ± 8.1 *
CMJ Post-Re-WU	43.1 ± 9.0	42.6 ±8.6	43.0 ±8.4	42.8 ± 8.6
Hamstring flexibility (°)				
Pre	90.2 ± 15.9	87.9 ± 17.0	89.7 ± 16.4	89.0 ± 16.6
Post	91.1 ± 15.8	94.4 ± 18.40 *	92.0 ± 17.0	92.6 ± 17.5 *

Mean values ± SD; CMJ, countermovement jump; CMJ Post-WU, CMJ after the warm-up; CMJ Post-S, CMJ after the stretching procedure; CMJ Post-Re-WU, CMJ after the re-warm-up; CTRL, control group; CRAC, contract-relax with antagonist contraction; DS, dynamic stretching; SS, static stretching. *: significant differences compared to baseline values (*p* < 0.05).

## Data Availability

All data files are available from a public database with the following https://doi.org/10.17632/3znpm68tyy.1 (https://data.mendeley.com/datasets/3znpm68tyy/1, accessed on 15 March 2025).

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
