# Peer review of "Acute Effects of Short Static, Dynamic, and Contract–Relax with Antagonist Contraction Stretch Modalities on Vertical Jump Height and Flexibility"

_sports, 2025, doi:10.3390/sports13040115_

Round 1
Reviewer 1 Report
Comments and Suggestions for Authors
Evaluation of manuscript sports-3540703
This study evaluated the effectiveness of 3 types of stretching on vertical jump. I will present below my suggestions to the authors.
The title is generic. I suggest a specific how: Dynamic stretching results in…
Add numerical results along with the statistics in the Abstract.
I really liked the introduction, however, there is one aspect that caught my attention: the main study used to justify the originality of this protocol was published in 2001.
Church, J.B.; Wiggins, M.S.; Moode, F.M.; Crist, R. Effect of Warm-up and Flexibility Treatments on Vertical Jump Performance. Journal of strength and conditioning research 2001, 15, 332–336.
However, when searching the literature, other recently published studies are found that compared different types of stretching.
Hammami, A., Selmi, W., Mahmoudi, A., Negra, Y., Chaouachi, A., Behm, D. G., ... & Hammami, R. (2025). Effects of neuromuscular versus stretching training performed during the warm-up on measures of physical fitness and mental well-being in highly-trained pubertal male soccer players. PloS one, 20(2), e0318318.
Chatzopoulos, D., Glynia, E., Symeonidis, M., Mouchou-Moutzouridou, E., Kanakari, P., Drakou, A., & Lykesas, G. (2025). Effects of Static Stretching, Dynamic Stretching, and Dance Warm-Up on Force Sense, Agility, and Attractiveness in Primary School Students. Journal of Exercise Physiology online, 28(IKEEART-2025-055), 44-55.
Flórez-Gil, E., Mateus, N., Sampaio, J., & Abade, E. (2025). Examining the impact of different re-warm-up strategies on non-starter basketball players’ physical performance. Journal of Sports Sciences, 1-8.
Therefore, I suggest that the authors base the original aspects of the present study on studies that have been recently published.
Insert the sample representativeness calculation and I believe it is important to present characteristics regarding the physical fitness level of the participants prior to data collection.
As for the methods, if the authors used a force platform, why did they only use jump height? Does the CMJ present many other interesting variables to be measured such as stiffness, peak force, rating of force development and flight time?
The Y axis in Figure 3 is incorrect, the correct one is jump height.
The discussion is well written, however, I miss the limitations of the present study, perspectives for future studies and practical applications. Please review.
Author Response
Reviewer #1 - Evaluation of manuscript sports-3540703
This study evaluated the effectiveness of 3 types of stretching on vertical jump. I will present below my suggestions to the authors.
Response to reviewer: Authors first thank the reviewer for the interesting and helpful comments. Please see below a point-by-point response to all of your comments. Alterations in the manuscript are presented in red font.
The title is generic. I suggest a specific how: Dynamic stretching results in…
Response to reviewer: Authors have not modified the title. The title proposed by the reviewer suggest the main stretching modality is dynamic stretching. It is not the case. Authors prefer the generic title initially submitted.
Add numerical results along with the statistics in the Abstract.
Response to reviewer: Values have been added in Abstract.
I really liked the introduction, however, there is one aspect that caught my attention: the main study used to justify the originality of this protocol was published in 2001.
Church, J.B.; Wiggins, M.S.; Moode, F.M.; Crist, R. Effect of Warm-up and Flexibility Treatments on Vertical Jump Performance. Journal of strength and conditioning research 2001, 15, 332–336.
However, when searching the literature, other recently published studies are found that compared different types of stretching.
Hammami, A., Selmi, W., Mahmoudi, A., Negra, Y., Chaouachi, A., Behm, D. G., ... & Hammami, R. (2025). Effects of neuromuscular versus stretching training performed during the warm-up on measures of physical fitness and mental well-being in highly-trained pubertal male soccer players. PloS one, 20(2), e0318318.
Chatzopoulos, D., Glynia, E., Symeonidis, M., Mouchou-Moutzouridou, E., Kanakari, P., Drakou, A., & Lykesas, G. (2025). Effects of Static Stretching, Dynamic Stretching, and Dance Warm-Up on Force Sense, Agility, and Attractiveness in Primary School Students. Journal of Exercise Physiology online, 28(IKEEART-2025-055), 44-55.
Flórez-Gil, E., Mateus, N., Sampaio, J., & Abade, E. (2025). Examining the impact of different re-warm-up strategies on non-starter basketball players’ physical performance. Journal of Sports Sciences, 1-8.
Therefore, I suggest that the authors base the original aspects of the present study on studies that have been recently published.
Response to reviewer: Authors thank the reviewer for this comment. Authors agree that some other studies compared different types of stretching. From the 3 proposed by the reviewer only one compared different stretching modalities. The two others (Hammami et al. (2025) compared a dynamic stretching to a neuromuscular training and Florez-Gil et al. (2025) compared dynamic stretching to plyometric activities). The last one (Chatzopoulos et al., 2025) compared static to dynamic stretching (authors added this reference in introduction). Authors are aware about the numerous studies that have compared these two stretching modalities. However, in this manuscript, authors used a study published in 2001 to justify the rationale since, to the best of authors’ knowledge, studies using CRAC and comparing CRAC to other stretching modalities for jump performance and flexibility are very scarce. Expect adding a reference in introduction, no other alteration was made.
Insert the sample representativeness calculation and I believe it is important to present characteristics regarding the physical fitness level of the participants prior to data collection.
Response to reviewer: An a priori sample calculation was done. The details were added in Methods. In addition, the physical fitness level was added. As for the methods, if the authors used a force platform, why did they only use jump height? Does the CMJ present many other interesting variables to be measured such as stiffness, peak force, rating of force development and flight time?
Response to reviewer: Vertical jump height was quantified using a force plate. Authors decided to have a single measurement in order to have a main message as simple as possible. The fact that other parameters could have been quantified has been added as limitation/perspective in the discussion.
The Y axis in Figure 3 is incorrect, the correct one is jump height.
Response to reviewer: Authors disagree with reviewer. Figure 3 present vertical jump changes and not jump height. Values are in percentages. Figure caption also clearly indicate it is percent changes of the vertical jump height (CMJ). Accordingly, no alteration was made.
The discussion is well written, however, I miss the limitations of the present study, perspectives for future studies and practical applications. Please review.
Response to reviewer: Authors added a limitation/perspective section in the discussion.
Reviewer 2 Report
Comments and Suggestions for Authors
The objective of the article is: The present study investigated the acute effects of different stretching modalities applied within a warm-up on flexibility and vertical jump height.
The authors are requested to address the following items:
- It is unclear whether the studied sample (n = 37) is sufficient to achieve reliable correlations. It is recommended to use software such as G*Power to justify the adequacy of the sample size.
- The authors do not demonstrate that the sample is representative of the population. Therefore, the study design cannot be classified as "experimental" (Subsection 2.2). It is recommended to change the classification to quasi-experimental.
- The study must specify its strengths and limitations in the Discussion section.
- The study must outline new future research directions derived from the obtained results (Discussion section).
Author Response
Reviewer #2
The objective of the article is: The present study investigated the acute effects of different stretching modalities applied within a warm-up on flexibility and vertical jump height.
Response to reviewer: Authors first thank the reviewer for the interesting and helpful comments. Please see below a point-by-point response to all of your comments. Alterations in the manuscript are presented in red font.
The authors are requested to address the following items:
It is unclear whether the studied sample (n = 37) is sufficient to achieve reliable correlations. It is recommended to use software such as G*Power to justify the adequacy of the sample size.
Response to reviewer: An a priori sample calculation was done before ongoing the present experiment. G*Power revealed a minimum of 32 volunteers to obtain a medium effect. The details were added in Methods.
The authors do not demonstrate that the sample is representative of the population.
Response to reviewer: The characteristics of the volunteers (physically active) have been added in Methods.
Therefore, the study design cannot be classified as "experimental" (Subsection 2.2). It is recommended to change the classification to quasi-experimental.
Response to reviewer: All volunteers performed all four experimental conditions. Conditions were randomly presented to the volunteers. Accordingly, the design is experimental and not quasi-experimental.
The study must specify its strengths and limitations in the Discussion section.
The study must outline new future research directions derived from the obtained results (Discussion section).
Response to reviewer: Authors added a limitation/perspective section at the end of the discussion.
Reviewer 3 Report
Comments and Suggestions for Authors
Dear Authors,
Overall a simple straight forward study looking at different modalities on vertical jump and flexibility. The inclusion of CRAC is novel. The methods, results and conclusions are all aligned with the purpose. My main question is flexibility was only addressed in the hamstring not the other two muscle groups that were stretched. There was no test of the quadriceps or planter flexors in the study. Research has shown that these two muscle groups are the big drivers of vertical jump performance as it is a plantar flexor and knee extensor movement, while the hamstring is a minor hip extensor. Why was the flexibility of these not measured?
Author Response
Reviewer #3
Dear Authors,
Overall a simple straight forward study looking at different modalities on vertical jump and flexibility. The inclusion of CRAC is novel. The methods, results and conclusions are all aligned with the purpose. My main question is flexibility was only addressed in the hamstring not the other two muscle groups that were stretched. There was no test of the quadriceps or planter flexors in the study. Research has shown that these two muscle groups are the big drivers of vertical jump performance as it is a plantar flexor and knee extensor movement, while the hamstring is a minor hip extensor. Why was the flexibility of these not measured?
Response to reviewer: Authors thank the reviewer for this comment. The present study explored stretching effects on vertical jump and flexibility. Vertical jump is mostly influence by quadriceps and plantar flexors than hamstring muscles. Because these 3 muscle groups were stretched here, the vertical jump test is justified. In contrast flexibility was only quantified in hamstring muscles for simplicity (authors wanted to make the findings clear for the reader and to give a final message as simple as possible). This is a limitation that is now addressed in the discussion. However, authors took care not to be speculative and to avoid making a direct link between our flexibility results and our vertical jump findings.
Round 2
Reviewer 1 Report
Comments and Suggestions for Authors
The authors have made an effort to improve the current version, even in points where I was not met, I feel honored by the authors' justifications, therefore, I consider the current text suitable for publication in Sports.